# Self-inhibiting percolation and viral spreading in epithelial tissue

**Xiaochan Xu[1,2], Bjarke Frost Nielsen[3,4], Kim Sneppen[1]***

[1]Niels Bohr Institute, University of Copenhagen, Copenhagen, Denmark; [2]Novo Nordisk Foundation Center for Stem Cell Medicine, reNEW, University of Copenhagen, Copenhagen, Denmark; [3]PandemiX Center, Department of Science and Environment, Roskilde University, Roskilde, Denmark; [4]High Meadows Environmental Institute, Princeton University, Princeton, United States

**Abstract** SARS-CoV-2 induces delayed type-I/III interferon production, allowing it to escape the early innate immune response. The delay has been attributed to a deficiency in the ability of cells to sense viral replication upon infection, which in turn hampers activation of the antiviral state in bystander cells. Here, we introduce a cellular automaton model to investigate the spatiotemporal spreading of viral infection as a function of virus and host-dependent parameters. The model suggests that the considerable person-to-person heterogeneity in SARS-CoV-2 infections is a consequence of high sensitivity to slight variations in biological parameters near a critical threshold. It further suggests that within-host viral proliferation can be curtailed by the presence of remarkably few cells that are primed for IFN production. Thus, the observed heterogeneity in defense readiness of cells reflects a remarkably cost-efficient strategy for protection.

*For correspondence:
sneppen@nbi.ku.dk

Competing interest: The authors declare that no competing interests exist.

## eLife assessment

This study presents a cellular automaton model to study the dynamics of virus-induced signalling and innate host defense against viruses such as SARS-CoV-2 in epithelial tissue. The simulations and data analysis are **convincing** and represent a **valuable** contribution that would be of interest to researchers studying the dynamics of viral propagation.

## Introduction

Adaptive immune responses are relatively slow since they require pathogen-specific priming of immune cells (*Sette and Crotty, 2021*). For example, the time required for the body to activate adaptive immunity against the SARS-CoV-2 virus upon initial infection is around 10 days, comparable to the delay of immunization against SARS-CoV-2 after vaccination (*Polack et al., 2020*). Instead, the earliest infection dynamics are largely governed locally, by infected cells and their neighborhood. The innate responses including both interferon (IFN) mediated intercellular communication and expression of antiviral genes (ISGs) are determinants for confining the viral spread in the respiratory tract. Here, we address the spread of viruses within epithelial tissue, using SARS-CoV-2 as a model pathogen. The overall considerations are similar for other viruses, but the parameters governing infection may vary considerably due to the specific countermeasures of the virus in question, affecting its ability to bypass human antiviral defenses.

In terms of countermeasures, insufficient type I and III interferon secretion upon infection is a main immune signature feature of SARS-CoV-2 infection (*Blanco-Melo et al., 2020*; *Hatton et al., 2021*; *Stanifer et al., 2020*; *Minkoff and tenOever, 2023*). The failure to activate immediate antiviral responses with IFNs is also a pathogenic aspect of other viruses including Ebola (*Mohamadzadeh*

*et al., 2007*), Marburg (*He et al., 2019*), and Herpes simplex (*Barreca and O'Hare, 2004*). Secretion of IFN relies on the cell's ability to sense viral products during its replication. Despite the presence of sensors for DNA and RNA viruses in cells, many species of viruses partially evade detection. The SARS-CoV-2 virus is such a case: Only two of 16 putative RNA virus sensors, IFIH1 (MDA5) and DHX58 (LGP2) from the RIG-I-like receptor (RLR) family, play roles in inducing IFN upon SARS-CoV-2 infection (*Yin et al., 2021*) and IFIH1 is antagonized by SARS-CoV-2 (*Liu et al., 2021*).

Intriguingly, evidence shows that pre-activated innate immune states help combat the SARS-CoV-2 infection. The higher basal expression of viral sensors, IFIH1 and DDX58 (also from the RLR family), in the upper airway of children (relative to adults), reduces the severity of COVID (*Loske et al., 2022*). Furthermore, well-differentiated primary nasal epithelial cells derived from a donor with pre-activated IFNγ show resistance to SARS-CoV-2 infection (*Broadbent et al., 2022*). Thus, the extent to which innate immunity contributes to the observed heterogeneity in responses to SARS-CoV-2 between hosts (*Schaller et al., 2021*; *Desai et al., 2020*) is a compelling subject for investigation.

To address this question, we reanalyze single-cell RNAseq data (*Fiege et al., 2021*; *Ravindra et al., 2021*) providing gene expression profiles of virus sensors and antiviral genes in host cells during early SARS-CoV-2 infection. We propose a cellular automaton model based on a few transition rules suggested by observed cell states, to explain the heterogeneity in early disease progression as a consequence of criticality in the virus-host interaction system.

## Results
### Cell states during early infection

Directly observing cell responses and cell state transitions in a patient's body upon viral infection is virtually impossible. Human bronchial epithelial cells (HBECs) mimic the airway epithelium and have been used as a representative model for investigating the consequences of the viral invasion (*de Jong et al., 1994*; *de Jong et al., 1993*; *Davis et al., 2015*). Single-cell RNAseq provides snapshots of the states of individual cells indicated by high-dimensional gene expression profiles at the mRNA level and can uncover the heterogeneity of cell responses obscured by aggregate measurement. Thus, by combining HBECs as a model and single-cell RNAseq data, one can in principle infer cell state transitions following viral infection. More importantly, single-cell RNAseq also captures copies of viral genes during sequencing, which allows us to estimate viral replication inside cells simultaneously.

To reconstruct the trajectory of cell state transitions during early SARS-CoV-2 infection, we reanalyze single-cell RNAseq data from experiments where HBECs are sampled from different conditions: Mock (corresponding to state before infection, 0 hr), 24 and 48 hours post-viral infection (hpi) (*Fiege et al., 2021*). We focus on genes associated with antiviral responses, interferon genes from the host cells, and detected viral genes. We project high-dimensional gene expression data onto a 2D plane using Uniform Manifold Approximation and Projection (UMAP) and obtain a low-dimensional visualization of single-cell expression patterns (*Figure 1a*). As a dimension reduction algorithm, UMAP is a manifold learning technique that favors the preservation of local distances over global distances (*McInnes et al., 2018*; *Becht et al., 2019*). It constructs a weighted graph from the data points and optimizes the graph layout in the low-dimensional space. On the UMAP plane (*Figure 1a*), each dot represents a cell sample and the distance between dots correlates with the level of similarity of cellular states. The cells are not divided absolutely into discrete clusters and rather show continuous trajectories. We cluster the cells with the principal components analysis (PCA) results from their gene expression. With the first 16 principal components, we calculate k-nearest neighbors and construct the shared nearest neighbor graph of the cells then optimize the modularity function to determine clusters. We present the cluster information on the UMAP plane and use the same UMAP coordinates for all the plots in this paper hereafter.

Different clusters on the UMAP indicate distinct cellular states during the progression of infection. For instance, there are three sub-clusters of susceptible cells ($O_1, O_2, O_3$). Neither viral genes nor IFNs are detected in these cells and only a few antiviral genes are expressed. The viral sensors (DHX58, DDX58, and IFIH1) are at their lowest level (*Figure 1b*, *Figure 1—figure supplement 1*). We refer to all of these cells as $O$ cells due to their relatively similar gene expression profiles in terms of viral replication genes. The proportion of $O$ cells decreases over time as the infection spreads (*Figure 1c*).

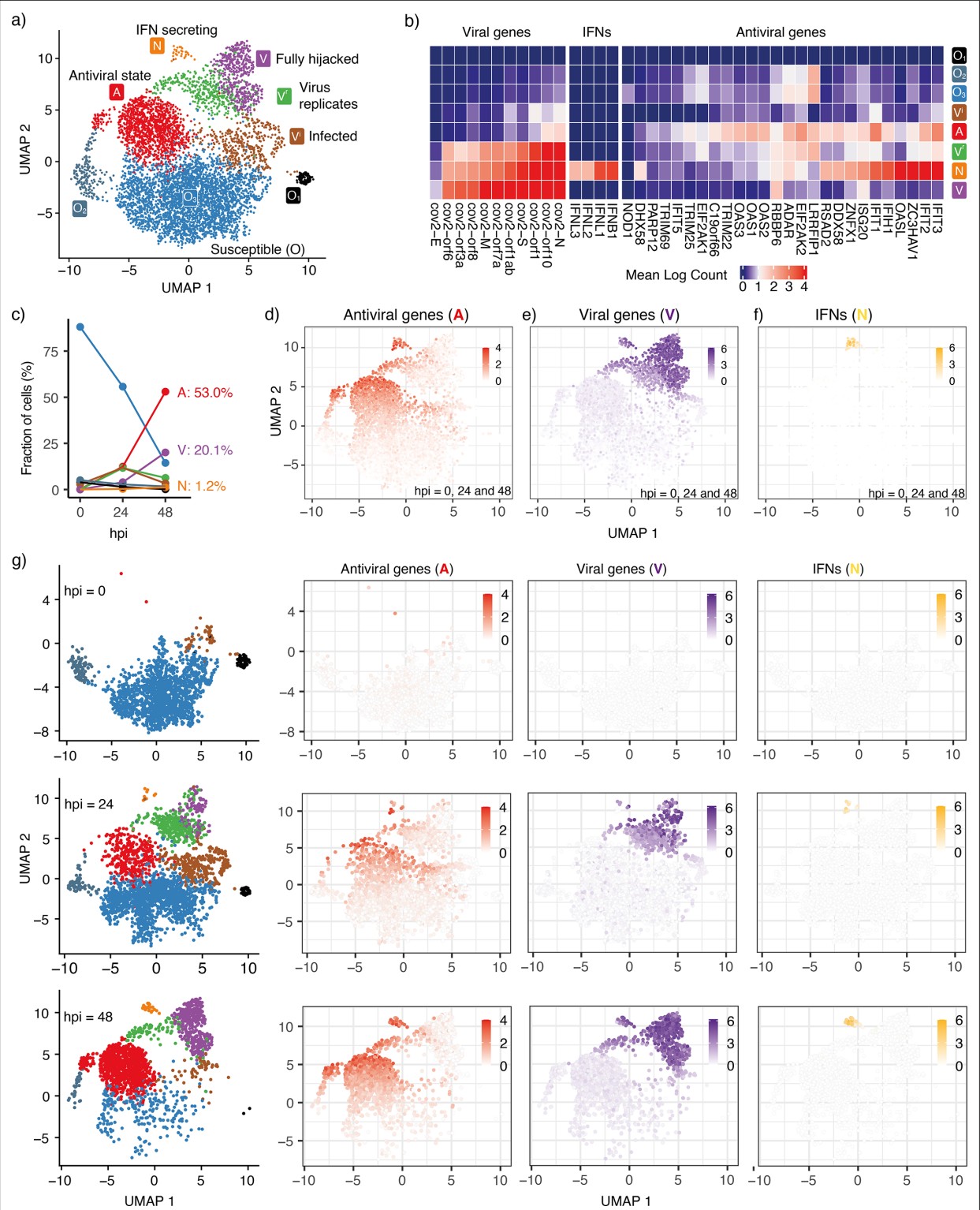

**Figure 1.** Cell states during SARS-CoV-2 infection in human tracheal/bronchial epithelial cells. (**a**) 6162 cells (**Fiege et al., 2021**) covering samples of mock-infected (0 h), 24 hpi (hours post-infection), and 48 hpi visualized with UMAP. (**b**) Average expression of representative viral genes, IFNs, and antiviral genes in cells within each cluster (state). (**c**) Cell proportions of clusters at different time points (hpi = 0, 24, and 48). The same colors are used for the lines as for the cluster (state) in (**a**). (**d**) Average expression of antiviral genes (IFIT1, IFIT2, IFIT3, IFIT5, IFIH1, OAS1, OAS2, OAS3, OASL, DDX58) in each cell. (**e**) Average expression of viral genes (cov.orf1ab, cov.S, cov.orf3a, cov.orf6, cov.M, cov.N) in each cell. (**f**) Average expression of interferon genes (IFNB1, IFNL1, IFNL2, IFNL3) in each cell. 103 cells (1.7%) are IFN-positive. (**g**) Progression of viral infection as indicated by changes in cell

*Figure 1 continued on next page*

*Figure 1 continued*

proportions of different states. Cells are shown separately at each time point in the leftmost column. The right columns show the average expression of antiviral genes, viral genes, and IFNs in each of these cells. Colorkeys indicate the gene expression level from low (white) to high (red, purple, or yellow).

The online version of this article includes the following figure supplement(s) for figure 1:

**Figure supplement 1.** Expression of antiviral genes at different stages of early SARS-CoV-2 infection.

**Figure supplement 2.** Detected viral genes and expression of IFNs at different stages of early SARS-CoV-2 infection.

We also observe three infected cell clusters where viral genes are primarily detected, $V^i$, $V^r$, and $V$. With the increasing counts of viral genes, we infer that the $V^i$ cluster is the earliest state after an $O$ cell has been infected and the virus begins replicating. Some but not all antiviral genes are activated in the $V^i$ cells (IFIT1/2/3 and OAS1/2/3; *Figure 1b* and *Figure 1—figure supplement 1*), indicating that these cells are still vulnerable to viral invasion. This cluster is followed by two subsequent clusters, the $V^r$ cluster with pronounced viral replication and $A$ cluster with barely any viral replication.

In the $V$ cluster, the viral genes reach their highest level, and antiviral genes are strongly inhibited, indicating that the virus has fully hijacked the cell. The antiviral genes are expressed most strongly in the $A$ cluster and partially in the $N$ cluster, indicating that the antiviral capability of the $N$ cluster is weaker than the full antiviral state. Although the $N$ cluster also shows a high level of viral genes, it severely lacks one of the viral genes (cov.E, *Figure 1—figure supplement 2*) compared with the most highly expressed viral genes of the $V$ cluster. This observation implies that viral replication and activation of the antiviral state coexist in the IFN-secreting cells ($N$ cluster). We note the existence of a small subgroup of the $V^r$ cluster, close to the $A$ cluster, that exhibits relatively high levels of both antiviral genes and viral genes but no appreciable IFN (*Figure 1d–f*). As in the $N$ cluster, the viral gene E is barely detected in these cells, indicating incomplete viral replication. However, in contrast to the $N$ cluster, the antiviral genes are expressed to their full extent (*Figure 1—figure supplements 1 and 2*). Thus, these cells are more likely to sustain the antiviral state.

At 24 hpi, some cells have switched from the pre-infection state ($O$) to other states. At 48 hpi, almost all cells have transitioned to other states and only a few cells remain in the $O$ state (*Figure 1c and g*). The aggregated gene expression of representative antiviral genes and detected viral genes indicates the cells move from the $O$ state towards the three remaining terminal states on the considered timescale of 2 days: Antiviral state ($A$, *Figure 1d*), Virus-conquered state ($V$, *Figure 1e*), and IFN producing state ($N$, *Figure 1f*). Central for the overall defense is the relatively few cells that reach the IFN-producing state ($N$). These cells also express $A$ and $V$ genes.

When IFN is not expressed, the antiviral genes and viral genes exclude each other (*Figure 1d and e*), except for a few cells around $(\mathrm{UMAP1}, \mathrm{UMAP2}) \sim (-2.5, 7.5)$ (green cells at hpi = 48, *Figure 1g*). They represent cells where the virus succeeded in stopping IFN secretion, but could not fully hijack the cell. We still regard these cells as antiviral cells in our model.

The $N$ state is associated with both high levels of virus sensors and viral genes, in agreement with the observation that IFN production is initiated after exposure to the virus (*Lei et al., 2020*) and that IFN can induce an antiviral state inside the same cell (*Sancéau et al., 1987*). Expression of the key SARS-CoV-2 sensitive sensors (IFIH1, DDX58, DHX58) is sparse in the $O$ state (*Figure 1—figure supplement 1*), indicating that a small fraction of cells have virus-sensing capacity prior to infection and are ready to mount a defense. This cell population increases with IFN tissue diffusion.

## Cellular automaton model capturing the cell state dynamics

We introduce a cellular automaton model to capture the cell state dynamics during the early stages of SARS-CoV-2 infection in a sheet of epithelial tissue. At each simulation, we seed an infection site on a 2D square lattice and study how the infection spreads as the sites on the lattice switch between cell states following a set of simple rules derived from the observations of the single-cell RNAseq data.

In addition to the states corresponding to the dominant clusters observed in the single-cell data (*Figure 1a*; $O$,$A$,$V$, and $N$ states corresponding to $O$, $A$, $V$, and $N$ clusters), we introduce a transient pre-antiviral state ($a$) that can switch to the $N$ state rapidly upon viral exposure, considering the heterogeneity of viral sensing ability in susceptible cells.

It follows from this description, that those RNA viruses that can be sensed by a larger repertoire of sensors should be modeled with a larger fraction of cells in the $a$ state.

The model is initialized with cells predominantly in the $O$ state and a small fraction, $p_a$, in the pre-antiviral state $a$.

Alternatively, one could formulate an equivalent model in which the initial state consisted entirely of $O$ cells (and an infection seed), and the parameter $p_a$ would instead be understood as the probability for an $O$ cell to switch to the $N$ or $A$ state when exposed to the virus or IFNs, respectively. This would be functionally equivalent to our model, and as such, the value of $p_a$ must depend on both host and virus. In particular, a virus that can effectively interfere with the defense and signaling of host cells will be modeled by a low $p_a$ value.

It is worth noting that the proportion of cells in the $a$ state before the onset of SARS-CoV-2 infection is expected to be higher in hosts with pre-activated antiviral innate immunity (*Loske et al., 2022*; *Broadbent et al., 2022*), meaning that the value of $p_a$ will, in general, depend on the exposure history of the host.

The cell state transitions triggered by IFN signaling or viral replication are known in viral infection, but how exactly the transitions are orchestrated for specific infections is poorly understood. The UMAP cell state distribution hints at possible preferred transitions between states. The closer two cell states are on the UMAP, the more likely transitions between them are, all else being equal. For instance, the antiviral state ($A$) is easily established from a susceptible cell ($O$), but not from the fully

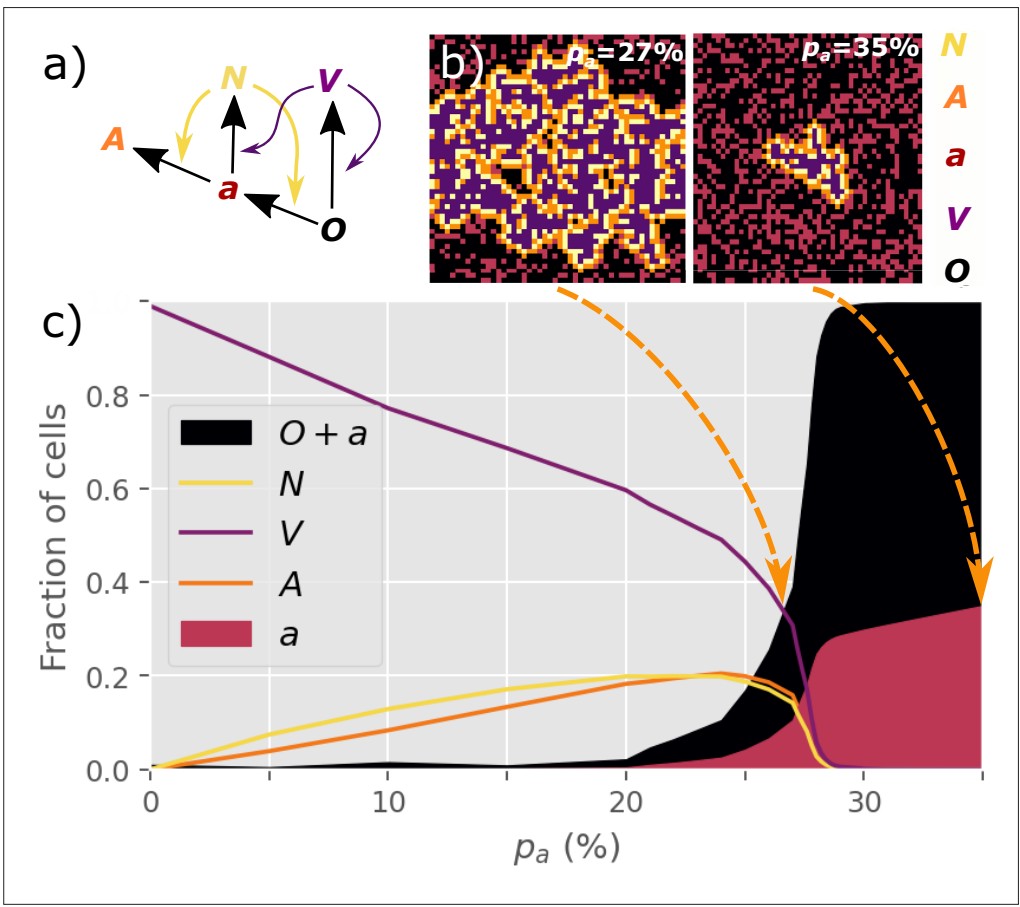

**Figure 2.** NOVAa model. (**a**) The cell state transitions are included in the NOVAa model. The straight black arrows indicate transitions between cell states. The curved yellow arrows indicate the effects of IFNs on activating antiviral states. The curved purple arrows indicate viral spread to cells with $O$ and $a$ states. (**b**) Final states of a small lattice (50 × 50) simulations at two different values of $p_a$ (both at IFN spreading radius $R = 1$). (**c**) The fraction of cells in each state in the final frozen configuration as a function of $p_a$. A critical transition is observed at $p_a = p_c \sim 27.8\%$. At lower values of $p_a$, most cells terminate in the $V$ state, representing an aggressive tissue infection. Simulations were performed on a lattice with linear dimension $L = 1000$.

The online version of this article includes the following figure supplement(s) for figure 2:

**Figure supplement 1.** Stochastic conversion.

virus-hijacked cell ($V$). The IFN-secreting cell state ($N$) requires the co-presence of the viral and anti-viral genes and thus the cell cluster is located between the antiviral state ($A$) and virus-infected state ($V$) but distant from the susceptible cells ($O$).

Inspired by the UMAP data visualization (**Figure 1a**), we propose the following transitions between five main discrete cell states (**Figure 2a**):

- $N$, IFN-secreting cells. These arise from pre-antiviral cells ($a$ state) that become infected (but not infectious). Here, we assume that the secretion of IFNs by the $N$ cells is a faster process than possible apoptosis (**Wen et al., 1997**; **Tesfaigzi, 2006**) of these cells and that the diffusion of IFNs to the neighborhood is not significantly affected by apoptosis.
- $O$, unaffected (susceptible) cells.
- $V$, infected and virus-producing cells. This state arises when a susceptible ($O$) cell is exposed to a virus from another $V$ cell.
- $a$, pre-antiviral state. It develops into either the $A$ or $N$ state upon exposure to signals from $N$ cells or virus from $V$ cells.
- $A$, antiviral state immune to infection. It is achieved when a pre-antiviral ($a$) cell is exposed to IFN. We do not consider the decay of the antiviral state as it may last more than 72 hr (**Gaajetaan et al., 2013**).

The dynamics are defined in terms of discrete time steps, representing the characteristic timescales of cellular viral infection. We explore the model for an extended time, keeping in mind that in reality other immune cells such as natural killer (NK) cells and macrophages may migrate to the infected site and reduce viral spread (**McNab et al., 2015**).

The four rules of the model are (**Figure 2a**):

$$N\left(a\right) = A, N\left(O\right) = a, V\left(a\right) = N, V\left(O\right) = V$$

where the notation $X\left(Y\right) = Z$ denotes a cell in state $X$ acting on a cell in state $Y$ and changing it to state $Z$ in one time-step. Thus, cells in states $O$, $a$, and $A$ are unable to influence their neighbors. The $V$ state is the only directly self-replicating state.

Each site of the $L \times L$ lattice is assigned to either the $O$ (probability: $1 - p_a$) or the $a$ state (probability: $p_a$). Infection is initiated by a single $V$ cell, and we explore the percolation of the infection to larger scales. A time step consists of $L^2$ updates, in which a random site $i$ is selected. If a $V$ cell is selected, it interacts with its 4 nearest neighbors according to the rules $V\left(O\right) = V$ and $V\left(a\right) = N$. If an $N$ cell is selected, it interacts with all cells within a radius $R$, according to the rules $N\left(O\right) = a$ and $N\left(a\right) = A$. The radius $R$ thus quantifies the diffusion range of IFNs relative to the virus. Periodic boundary conditions are imposed in the model throughout.

## Criticality in viral spreading

At $R = 1$, the final number of infected cells depends strongly on the value of $p_a$. At a low $p_a$ of 0.27, infections typically spread to the entire system, while at a higher $p_a$ of 0.35, the propagation of the $V$ state is inhibited (**Figure 2b**).

We observe a threshold-like behavior of the final attack rate of the virus when the initial $p_a$ changes continuously (**Figure 2c**). The virus spreads macroscopically for $p_a < p_c \approx 27.8\%$. At higher $p_a$, cells are sufficiently prone to convert to the antiviral state to prevent the infection from percolating. We explore a version of the dynamics where interactions only happen with a reduced probability $p_{conv} < 1$ rather than being deterministically applied to all neighbors (**Figure 2—figure supplement 1**). We find that this does not affect the critical behavior of the model.

The size distribution $P\left(s\right)$ of infection clusters (defined as the number of $N$ and $V$ cells in a cluster) around the critical value of $p_a = 27.8\%$ obeys power-law decay (**Figure 3a**). In **Figure 3b**, the distribution $P\left(s\right) \propto 1/s^\tau$ is further explored by re-scaling and the cluster size exponent is confirmed as $\tau = 1.83 \pm 0.03$ when $p_a = 28\%$. Notably, this exponent is below the equilibrium 2D percolation yielding $\tau = 2.05$ (**Stauffer and Aharony, 2018**). Further, our exponent $\tau \sim 1.8$ is above the percolation-inspired cluster growth model for virus spread (**Gönci et al., 2010**) which has an exponent between $\tau = 1.58$ and $\tau = 1.64$ depending on the distribution of individual cells' pre-defined ability to become infected. Meanwhile, the propagation for the different states could be accelerated by the smaller value of $p_a$ with the same $R$ (**Figure 3—figure supplement 1**).

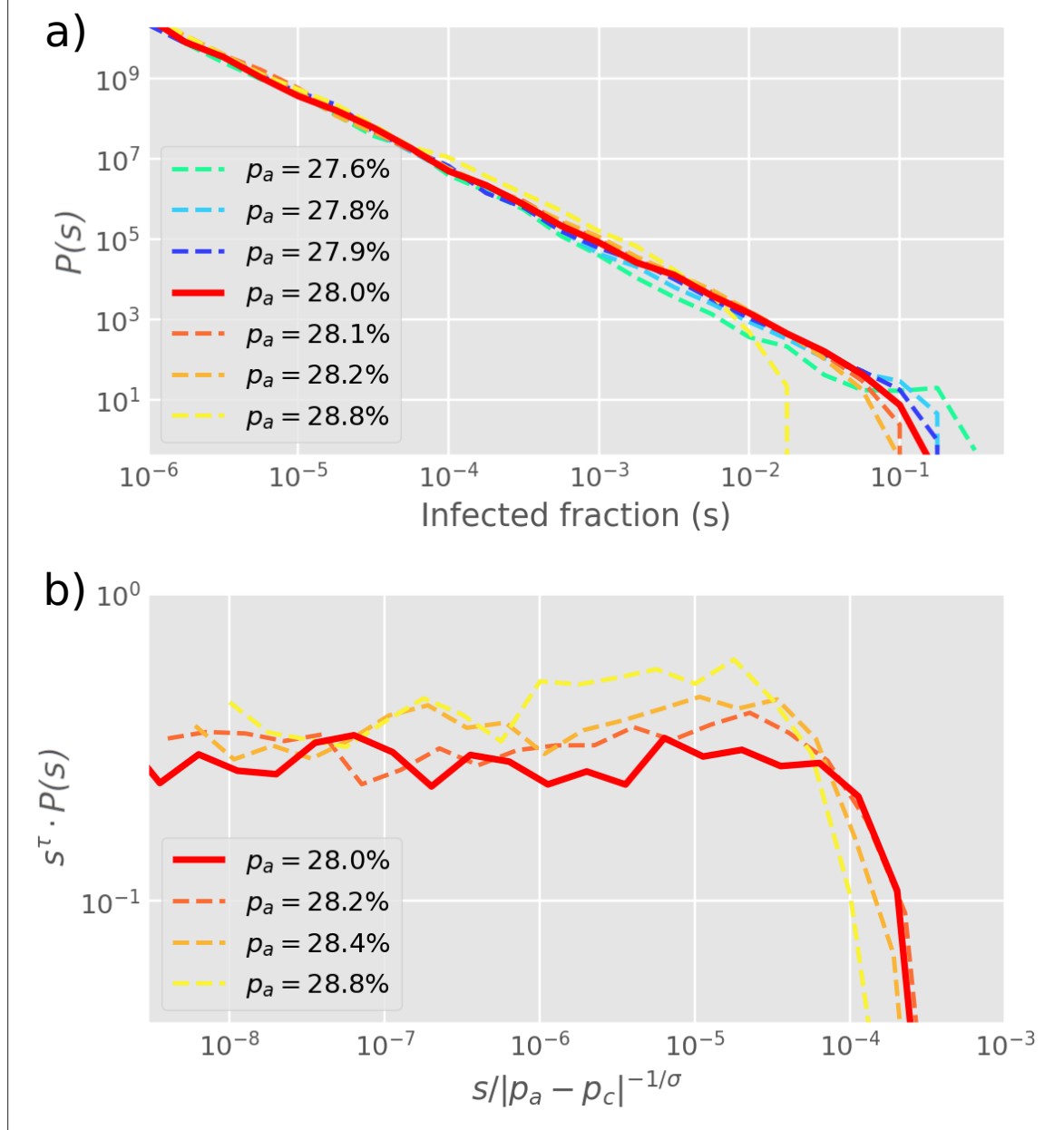

**Figure 3.** Cluster size distribution. (**a**) The distribution $P(s)$ of cluster sizes of infected cells ($s = (N + V)/L^2$) for different values of $p_a$, simulated by starting with one infected cell in a 2D square lattice of linear extent $L = 2000$. (**b**) The exponents from (**a**) are extracted by re-scaling $P(s)$ as shown on the $y$-axis, yielding $\tau = 1.83$. The cut-off exponent is estimated as $\sigma \sim 1$. Simulations plot the final outbreak sizes from 10,000 initial infections of one cell. The histogram is log-binned with 5 bins per decade. The critical point at $p_a = p_c = 0.28$ is determined as the value with the longest scaling regime.

The online version of this article includes the following figure supplement(s) for figure 3:

**Figure supplement 1.** Cell fractions within the different states over times, at a range of $p_a$ values (below the critical value $p_c$), for two interferon spreading radii, $R = 1$ and $R = 5$.

The actual critical value of $p_a$ depends strongly on the choice of neighborhood. In particular, at $R = 1$, the $V$ and $N$ states have the same range in the tissue (a proxy for diffusivity), while a more realistic scenario is to allow IFNs to diffuse faster in the tissue ($R > 1$), facilitating the initiation of the antiviral state. The critical percolation threshold $p_c$ decreases almost exponentially with the value of $R$ (**Figure 4b**), and viral propagation can be stopped for $p_a$ as low as 0.4% when $R \geq 5$. Such a small fraction of initial cells in the $a$ state is consistent with the remarkably few $N$ cells observed in

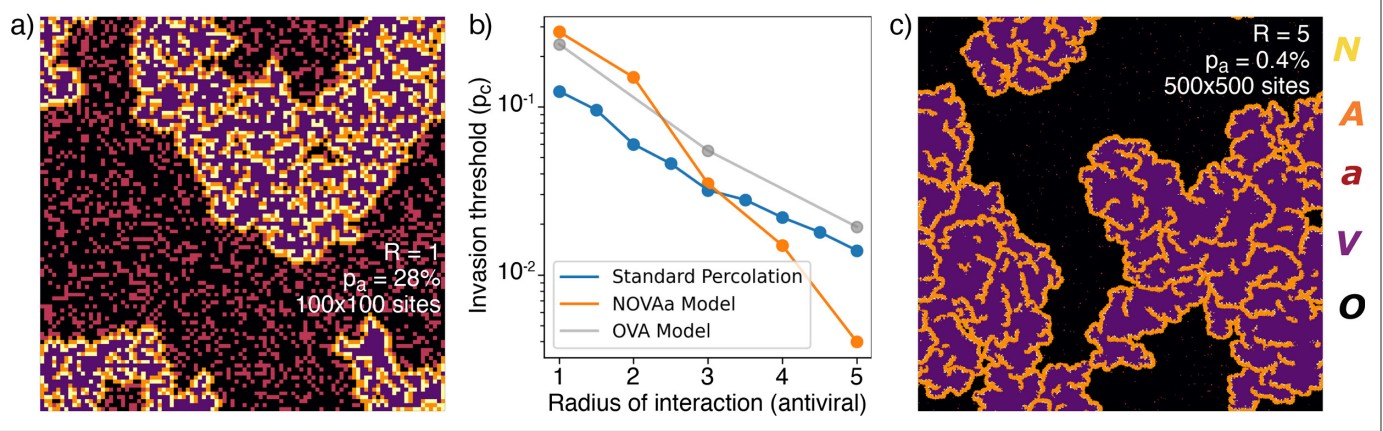

**Figure 4.** Range of IFN. (**a**) Typical cluster for an $R = 1$ simulation at $p_a \sim p_c = 0.278$. (**b**) The dependence of $p_c$ with $R$, approximately reproduced by a fit $p_c \sim 3^{-R}$. For comparison, the OVA model as well as percolation has $p_c \sim 3^{-R/2}$. In all cases, when $p_a$ is above $p_c$ then the virus is prevented from spreading. (**c**) Cluster distribution for $R = 5$ at $p_a \sim p_c = 0.004$, at a five times larger linear scale than (**a**).

The online version of this article includes the following figure supplement(s) for figure 4:

**Figure supplement 1.** We introduced the simplified OVA model, which does not include an $N$ state, but rather gives each susceptible cell a probability of spontaneously converting to an antiviral state in each time step.

**Figure supplement 2.** Comparison of the Gaussian Model and the main NOVAa model at identical parameters.

experiments (**Figure 1c**). Thus, a higher diffusivity of IFN provides a more than proportional decrease in the required number of antiviral cells. As revealed by the reanalysis of RNAseq data in **Figure 1**, the fraction of IFN-positive cells is relatively low – around 1.7%. Comparing with simulations near the critical point, we find that, at $R = 5$, the ratio of $N$ cells to all affected cells $(N + A + V)$ in the final state, $lim_{t \to \infty} N / (N + A + V) \approx 2\%$, i.e. it is of comparable magnitude to the experimental value. This holds in a wide range around the critical point, $p_a \sim p_c$.

The exponents for the cluster size distribution are the same at $R = 1$ and $R = 5$, while the structures of the clusters are different (**Figure 4a and c**). Greater $R$ leads to a different microscopic structure with fewer $A$ and $N$ cells in the final state (**Figure 4c**).

To put the above findings in perspective we further explore a simplified version of our model with only three states (**Figure 4—figure supplement 1**), the OVA model, which may be seen as a rephrasing of models for induced antiviral states suggested by **Howat et al., 2006**; **Segredo-Otero and Sanjuán, 2020**; **Michael Lavigne et al., 2021**. In the OVA model, $p_a$ is the probability that an infected cell produces interferons to warn neighbor cells within radius $R$. In the OVA model, one update consists of selecting a random cell. If the cell is in the $V$ state then its neighbor cells may change by exposure to the virus, provided that they are susceptible ($O$). Each of the four neighbors is now chosen in random order, and if a neighbor cell $i$ is in the $O$ state, a random number $ran_i \in [0, 1]$ is drawn. If $ran_i \geq p_a$ the neighbor is flipped to the $V$ state. If, on the other hand, $ran_i < p_a$, all $O$ cells within a radius $R$ around the neighbor $i$ are converted to the $A$ state. Thus, for large $R$ and moderate $p_a$, the spread of infection will be mitigated. We find that the OVA model has an 'outbreak size' exponent $\tau \sim 1.8$, similar to the NOVAa model. However, the change in microstructure as a function of the IFN range $R$ observed in the NOVAa model (compare **Figure 4a and c**) is not observed in the OVA model (**Figure 4—figure supplement 1**), where the features instead scale proportionally with $R$. We also simulated standard percolation by randomly adding disks of radius $R$ of blocking ('antiviral') cells and checking for percolation of the infected state. While the critical behavior of the standard percolation model approximately resembles that of the OVA model (**Figure 4b**), the antiviral state of the OVA model is somewhat less effective at blocking the spread (reflected in a higher threshold $p_c$).

Finally, we examined a version of the model where the discrete idealization of $N$ cells acting at all cells within a specific radius $R$ is replaced by a probabilistic conversion with a diffusion-like profile. The algorithm for this is described in the Methods, with results in **Figure 4—figure supplement 2** to be compared to **Figure 4**. We find that the probabilistic spreading of IFN is more effective, in terms of demanding lower $R$ for obtaining similar limits on the spreading of the infection. This is likely due

to the existence of long-range interactions (however rare) when neighbors are selected according to a Gaussian profile.

## Discussion

There are some preexisting models of the interplay between virus, host cells and triggered immune responses, with an antiviral state triggered by IFN signaling from neighbor cells (*Graw and Perelson, 2016*). Cellular automaton models of infection dynamics in epithelial tissue were explored by *Howat et al., 2006*; *Segredo-Otero and Sanjuán, 2020*; *Michael Lavigne et al., 2021*, with the overall result that spreading depends on competition between the virus and an induced antiviral cell state. This competition is recapitulated in our model in terms of the two effective parameters $p_a$ and $R$. Our model emphasizes the threshold dynamics, with a critical transition between effective confinement and unhindered spread that depends sensitively on the details of the relevant cell states. In particular, the presence of the specialized IFN-producing $N$ cells allows for disease confinement at a much lower concentration of pre-antiviral cells (lower value of $p_a$) in the NOVAa model, than in the OVA model which lacks the $N$ state (*Figure 4b*). As a consequence of low $p_a$, the number of final $N$-state cells is also much lower.

The low concentration of ready-to-fight cells may seem perplexing, leading one to surmise that the organism could easily fight off an infection by only slightly increasing its investment in these primed cells. However, one should keep in mind that for example the human organism does indeed have ready-to-fight cells that can eliminate most foreign RNA, and only leave a few truly infectious viruses. As highlighted in the introduction, these select viruses often employ strategies to lower the $p_a$, for example by only being sensed by a small fraction of the RNA virus-sensitive receptors of our cells.

The parameter $p_a$ can be interpreted as the probability that a cell is sufficiently antiviral to convert to a $N$ state upon infection with a given virus. The relevant value of $p_a$ will depend on the virus considered (and will be small for viruses that inhibit cell responses to infection) as well as on the host (e.g. on age [*Kissler et al., 2020*] and recent infection history). Dysregulated IFN responses are characteristic of the effective immunomodulatory strategies used by betacoronaviruses (*Channappanavar et al., 2019*; *Acharya et al., 2020*).

The parameter $R$ reflects the signaling efficiency of an interferon-producing cell. Since $R$ is measured in units of the typical distance that the virus spreads, it depends on viral properties, including its burst size, diffusion, and adsorption to host cells, with higher adsorption being associated with larger $R$ values. For SARS-CoV-2 this suggests that lower ACE2 receptor counts would result in less adsorption to nearby cells, in turn allowing the virus to spread to more distant tissues (*Bastolla, 2021*) suggesting a lower value of $R$.

For viruses that do not delay the production of IFNs, $p_a$ would be higher than for SARS-CoV-2, allowing neighbor cells around an infected site to form a kind of "ring vaccination" as the antiviral state dominates. In this sense, our model is consistent with the previous modeling of the roles of autocrine and paracrine interferon signaling suppression of viral infection (see e.g. *Michael Lavigne et al., 2021* for parallels between IFN response and 'ring vaccination').

We do not consider viral particles which enter the bloodstream and seed new infections non-locally. This may allow the virus to spread in the tissue at what would otherwise constitute sub-critical conditions in our model. Further, there may be tissue-specific variations in both $p_a$ and $R$, adding larger-scale heterogeneity to the overall spreading. As the disease progresses one would expect additional heterogeneity to emerge, associated with variability in later host responses including macrophage activation and adaptive immunity (*Wang et al., 2021*).

The remarkable heterogeneity of disease progression in COVID-19, in the form of widely variable symptoms (*Tabata et al., 2020*) and transmission risk (*Nielsen et al., 2021*; *Kirkegaard and Sneppen, 2021*), has been widely observed. For instance, among university students, just 2% of SARS-CoV-2 positive hosts provided 90% of total respiratory viral load (*Yang et al., 2021*). In our formalism, we would understand such variability in terms of a $p_a$ that is comparable to the critical value, but varying between hosts. A slight change of $p_a$ then results in dramatic fluctuations in the outcome of an infection.

To be more quantitative, for SARS-CoV-2 the detected virus count on average grows by a factor of 3.5 (*Kissler et al., 2020*) in one infection generation of 8 hr (not to be confused with the between-host 'generation time' of the infection). This within-host reproductive number is far below the number

of viruses produced from a cell, indicating severe restrictions from the innate immune system. On the other hand, 3.5 is still above the threshold for spreading, indicating that within-host amplification is super-critical. However, the measured amplification includes viruses that 'jump' to other spots in an infected person, thereby suggesting a local spreading that is closer to the critical value than an amplification of 3.5 would suggest.

Our study finally compared the NOVAa model with the simpler OVA scenario that recapitulates earlier modeling of induced antiviral states (*Howat et al., 2006*; *Segredo-Otero and Sanjuán, 2020*; *Michael Lavigne et al., 2021*). These papers all build on a more homogeneous role of infected cells, each inducing some immunization of surrounding cells. They emphasize the larger range of IFN signals compared to viral diffusion (*Howat et al., 2006*), focus on the competition between viruses with different abilities to suppress IFN signaling (*Segredo-Otero and Sanjuán, 2020*), or introduce a cellular automaton approach where the antiviral state leads to a type of ring vaccination that prevents the virus from spreading when more IFN is produced (*Michael Lavigne et al., 2021*). Our OVA model may be seen as a simplified and more stochastic version of the last model. The NOVAa model then adds the additional benefits associated with the experimentally observed but low-abundance $N$ state cells, which by their rarity adds to predicted randomness between the fate of individual infection centers during an early viral infection.

## Methods
### Stochastic conversion
While even the base model has a level of stochasticity – since $L^2$ are randomly chosen, with replacement, to be updated in each time step – we here simulate a version of the dynamics which includes stochastic conversion, that is each action of a cell on a neighboring cell occurs only with a probability $p_{conv}$ (and the original model is recovered as the $p_{conv} = 1$ scenario). This necessarily slows down the dynamics (or effectively rescales time by a factor $p_{conv}$), but crucially we find that it does not appreciably affect the location of the threshold $p_c$. In *Figure 2—figure supplement 1*, we show a parameter scan across $p_a$ values for $R = 1$ and $p_{conv} = 0.5$, which shows that the threshold continues to exist at around $p_a = 27\%$.

### Time-evolution of state occupancy
In *Figure 3—figure supplement 1*, we show the time-evolution of occupation fractions for the different states of the model, for various values of $p_a$ below the critical value $p_c$, for two interferon spreading radii, $R = 1$ and $R = 5$. Each panel is based on a single typical realization.

As shown qualitatively in the figure, the speed of propagation as well as the final occupancy ratios depend on the distance to the threshold, $|p_a - p_c|$.

### Simulations with a Gaussian kernel
In the NOVAa model of the main text, the spread of interferons (i.e. the action of cells in the $N$ state) always follows a circular motif. When an $N$ cell is selected, it will act on all cells within a radius $R$ (provided they are in the $O$ or $a$ state). To more closely approximate the diffusion of interferons – and to allow for some stochasticity in this process – we will here consider an extension of the model, in which the spread of interferon is modeled by a Gaussian kernel.

In the following, we will refer to the model presented in the main text as the *model with a circular spreading motif* and the alternative model as the *Gaussian model*.

The Gaussian model is implemented as follows:

Let

$$P\left(d; \sigma\right) = N \exp\left[-d^2 / \left(2\sigma^2\right)\right] \tag{1}$$

with $N$ a normalization constant and $\sigma = \frac{2\sqrt{2}}{3\sqrt{\pi}}R$. This value of $\sigma$ ensures that the mean distance to converted cells (i.e. those acted upon by $N$ cells) is the same as in the model with a circular spreading motif.

The normalizaton constant $N$ is chosen such that the average number of converted cells is the same as in the model with a circular spreading motif. This results in a value of $N = N_R / \left(2\pi\sigma^2\right)$ where $N_R$ is the number of lattice points within a radius of $R$ from a central lattice point.

The simulation routine then proceeds as follows:

- At each time step, $L^2$ random sites are selected for updating (with replacement).
- For non-$N$ cells, updates are carried out as in the main text.
- When an $N$ cell is selected, the update proceeds as follows:
  ○ All cells within a radius of $4\sigma$ are designated as neighbors.
  ○ For each of these neighbor cells, convert the cell (i.e. let the $N$ cell act on it) with probability $P\left(d;\sigma\right)$, where $d$ is the Euclidean distance between the neighbor cell and the $N$ cell.
  ○ Once all neighbours have been considered, move the $N$ cell to a new state $N_1$
- When an $N_1$ cell is selected, it behaves identically to an $N$ cell. Once an $N_1$ cell has been updated, it moves to state $N_2$, which is inactive.

The extended radius of $4\sigma$ was chosen to ensure that the vast majority of potential interactions are included, while retaining numerical efficiency.

The introduction of the two new (albeit very simple) states $N_1$ and $N_2$ was to ensure that a single $N$ cell does not act on a very large number of other cells simply by being selected multiple times during a simulation. In the circular motif case, this was automatic since an $N$ cell could only act on the same $N_R$ cells in each time step, and once they were in the $N$, $V$ or $A$ state, the $N$ cell could no longer affect them. In practice, an $N$ cell could act twice on a susceptible cell, once to turn it from $O$ to $a$ and once to convert it from $a$ to $A$.

In the Gaussian case, an $N$ cell could in principle act on an unlimited number of cells, although the rate would decrease with distance according to the Gaussian kernel. Thus, it was necessary to introduce some memory in the form of the $N_1$ and $N_2$ states to more closely mimic the circular motif case of the main text.

As shown in *Figure 4—figure supplement 2*, the Gaussian model can behave quite differently to the circular motif model even given the same parameter values. The primary reason for this difference owes to the nonzero probability of long-range conversions in the Gaussian model since this allows for bridging areas otherwise devoid of $a$ cells.

The C++source code for the simulations and a Python notebook for plotting can be found at: https://github.com/BjarkeFN/ViralPercolation (copy archived at *Bjarke, 2024*).

## Acknowledgements

BFN acknowledges financial support from the Carlsberg Foundation in the form of an Internationalisation Fellowship (grant no. CF23-0173) and through the Carlsberg Foundation's Semper Ardens programme (grant no. CF20-0046), as well as from the Danish National Research Foundation (grant no. DNRF170). Novo Nordisk Foundation Center for Stem Cell Medicine is supported by Novo Nordisk Foundation grant NNF21CC0073729.

## Additional information

### Funding

| Funder | Grant reference number | Author |
| --- | --- | --- |
| Carlsbergfondet | CF20-0046 | Bjarke Frost Nielsen |
| Novo Nordisk Fonden | NNF21CC0073729 | Xiaochan Xu |
| Carlsbergfondet | CF23-0173 | Bjarke Frost Nielsen |

| Funder | Grant reference number | Author |
| --- | --- | --- |

The funders had no role in study design, data collection and interpretation, or the decision to submit the work for publication.

## Author contributions

Xiaochan Xu, Conceptualization, Data curation, Software, Formal analysis, Investigation, Visualization, Methodology, Writing - original draft, Project administration, Writing - review and editing; Bjarke Frost Nielsen, Conceptualization, Data curation, Software, Formal analysis, Investigation, Visualization, Methodology, Writing - original draft, Writing - review and editing; Kim Sneppen, Conceptualization, Formal analysis, Supervision, Investigation, Visualization, Methodology, Writing - original draft, Writing - review and editing

## Author ORCIDs

Kim Sneppen ⬤ http://orcid.org/0000-0001-9820-3567

Reviewer #1 (Public Review): https://doi.org/10.7554/eLife.94056.3.sa1
Reviewer #2 (Public Review): https://doi.org/10.7554/eLife.94056.3.sa2
Reviewer #3 (Public Review): https://doi.org/10.7554/eLife.94056.3.sa3
Author response https://doi.org/10.7554/eLife.94056.3.sa4

# Additional files

## Supplementary files

• MDAR checklist

## Data availability

The data on gene expressions was obtained from a referenced publication (GSE157526). The modeling code is deposited on GitHub https://github.com/BjarkeFN/ViralPercolation (copy archived at *Bjarke, 2024*).

The following previously published dataset was used:

| Author(s) | Year | Dataset title | Dataset URL | Database and Identifier |
| --- | --- | --- | --- | --- |
| Jessica KF | 2020 | Single cell resolution of SARS-CoV-2 tropism, antiviral responses, and susceptibility to therapies in primary human airway epithelium | https://www.ncbi.nlm.nih.gov/geo/query/acc.cgi?acc=GSE157526 | NCBI Gene Expression Omnibus, GSE157526 |

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
