## [Editor Report · eLife assessment]

This study presents a cellular automaton model to study the dynamics of virus-induced signalling and innate host defense against viruses such as SARS-CoV-2 in epithelial tissue. The simulations and data analysis are **convincing** and represent a **valuable** contribution that would be of interest to researchers studying the dynamics of viral propagation.

---

## [Referee Report · Reviewer #1 (Public Review)]

Summary:

The manuscript describes a model based on 5-state cellular automata of development of an infection. The model is motivated and qualitatively justified by time-resolved measurements of expression levels of viral, interferon-producing, and antiviral genes. The model is set up in such a way that the crucial difference in outcomes (infection spreading vs. confinement) depends on the initial fraction of special virus-sensing cells. Those cells (denoted as 'type a') cannot be infected and do not support the propagation of infection, but rather inhibit it in a somewhat autocatalytic way. Presumably, such feedback makes the transition between two outcomes very sharp: a minor variation in concentration of 'a' cells results in qualitative change from one outcome to another. As in any percolation-like system, the transition between propagation and inhibition of infection goes through a critical state with all its attributes, including a power-law distribution of the cluster size (corresponding to the fraction of infected cells) with a fairly universal exponent and a cutoff at the upper limit of this distribution.

Strengths:

The proposed model suggests a well-justified explanation for the frequently observed yet puzzling diversity of outcomes of viral infections such as COVID.

Weaknesses:

None.

---

## [Referee Report · Reviewer #2 (Public Review)]

Xu et al. introduce a cellular automaton model to investigate the spatiotemporal spreading of viral infection. In this study, the author first analyzes the single-cell RNA sequencing data from experiments and identifies four clusters of cells at 48 hours post-viral infection, including susceptible cells (O), infected cells (V), IFN-secreting cells (N), and antiviral cells (A). Next, a cellular automaton model (NOVAa model) is introduced by assuming the existence of a transient pre-antiviral state (a). The model consists of an LxL lattice; each site represents one cell. The cells change their state following the rules depending on the interaction of neighboring cells. The model introduces a key parameter, p_a, representing the fraction of pre-antiviral state cells. Cell apoptosis is omitted in the model. Model simulations show a threshold-like behavior of the final attack rate of the virus when p_a changes continuously. There is a critical value p_c, so that when p_a < p_c, infections typically spread to the entire system, while at a higher p_a > p_c, the propagation of the infected state is inhibited. Moreover, the radius R that quantifies the diffusion range of N cells may affect the critical value p_c; a larger R yields a smaller value of the critical value p_c. The authors further examine the result with stochastic version dynamics, and the main findings are unchanged upon stochastic dynamics. The structure of clusters is different for different values of R; greater R leads to a different microscopic structure with fewer A and N cells in the final state. Compared with the single-cell RNA seq data, which implies a low fraction of IFN-positive cells of around 1.7%, the model simulation suggests R=5. The authors also explored a simplified version of the model, the OVA model, with only three states. The OVA model also has an outbreak size. The OVA model shows dynamics similar to the NOVAa model. However, the change in microstructure as a function of the IFN range R observed in the NOVAa model is not observed in the OVA model.

---

## [Referee Report · Reviewer #3 (Public Review)]

Summary:

This study considers how to model distinct host cell states that correspond to different stages of a viral infection: from naïve and susceptible cells to infected cells and a minority of important interferon-secreting cells that are the first line of defense against viral spread. The study first considers the distinct host cell states by analyzing previously published single-cell RNAseq data. Then an agent-based model on a square lattice is used to probe the dependence of the system on various parameters. Finally, a simplified version of the model is explored, and shown to have some similarity with the more complex model, yet lacks the dependence on the interferon range. By exploring these models one gains an intuitive understanding of the system, and the model may be used to generate hypotheses that could be tested experimentally, telling us "when to be surprised" if the biological system deviates from the model predictions.

Strengths:

- Clear presentation of the experimental findings and a clear logical progression from these experimental findings to the modeling.

- The modeling results are easy to understand, revealing interesting behavior and percolation-like features.

- The scaling results presented span several decades and are therefore compelling.

- The results presented suggest several interesting directions for theoretical follow-up work, as well as possible experiments to probe the system (e.g. by stimulating or blocking IFN secretion).

Weaknesses:

- The fixed time-step of the agent-based modeling may introduce biases. I would consider simulating the system with Gillespie dynamics where the reaction rates depend on the ambient system parameters.

- Single-cell RNAseq data requires careful handling or it may generate false leads. The strength of the RNAseq evidence presented is not clear.

Two places where the manuscript could be extended:

- Since the "range" of IFN is an important parameter, it makes sense to consider other lattice geometries other than the square lattice, which is somewhat pathological. Perhaps a hexagonal lattice would generalize better.

- Tissues are typically three-dimensional, not two-dimensional. (Epithelium is an exception). It would be interesting to see how the modeling translates to the three-dimensional case. Percolations transitions are known to be very sensitive to the dimensionality of the system.

Justification of claims and conclusions:

The claims and conclusions are well justified.

---

## [Author Response]

The following is the authors’ response to the original reviews.

**Public Reviews:**

**Reviewer #1 (Public Review):**
Summary:The manuscript "Self-inhibiting percolation and viral spreading in epithelial tissue" describes a model based on 5-state cellular automata of development of an infection. The model is motivated and qualitatively justified by time-resolved measurements of expression levels of viral, interferon-producing, and antiviral genes. The model is set up in such a way that the crucial difference in outcomes (infection spreading vs. confinement) depends on the initial fraction of special virus-sensing cells. Those cells (denoted as 'type a') cannot be infected and do not support the propagation of infection, but rather inhibit it in a somewhat autocatalytic way. Presumably, such feedback makes the transition between two outcomes very sharp: a minor variation in concentration of ``a' cells results in qualitative change from one outcome to another. As in any percolation-like system, the transition between propagation and inhibition of infection goes through a critical state with all its attributes. A power-law distribution of the cluster size (corresponding to the fraction of infected cells) with a fairly universal exponent and a cutoff at the upper limit of this distribution.Strengths:The proposed model suggests an explanation for the apparent diversity of outcomes of viral infections such as COVID.

Author response: We thank the referee for the concise and accurate summary of our work.

Weaknesses:Those are not real points of weakness, though I think addressing them would substantially improve the manuscript.

Author response: Below we will address these point by point.

The key point in the manuscript is the reduction of actual biochemical processes to the NOVAa rules. I think more could be said about it, be it referring to a set of well-known connections between expression states of cells and their reaction to infection or justifying it as an educated guess.

Author response: We have now improved this part in the model section. We have added a few sentences explaining how the cell state transitions are motivated by the UMAP results:

“The cell state transitions triggered by IFN signaling or viral replication are known in viral infection, but how exactly the transitions are orchestrated for specific infections is poorly understood. The UMAP cell state distribution hints at possible preferred transitions between states. The closer two cell states are on the UMAP, the more likely transitions between them are, all else being equal. For instance, the antiviral state (𝐴) is easily established from a susceptible cell (𝑂), but not from the fully virus-hijacked cell (𝑉 ). The IFN-secreting cell state (𝑁) requires the co-presence of the viral and antiviral genes and thus the cell cluster is located between the antiviral state (𝐴) and virus-infected state (𝑉 ) but distant from the susceptible cells (𝑂).

Inspired by the UMAP data visualization (Fig. 1a), we propose the following transitions between five main discrete cell states”

Another aspect where the manuscript could be improved would be to look a little beyond the strange and 'not-so-relevant for a biomedical audience' focus on the percolation critical state. While the presented calculation of the precise percolation threshold and the critical exponent confirm the numerical skills of the authors, the probability that an actual infected tissue is right at the threshold is negligible. So in addition to the critical properties, it would be interesting to learn about the system not exactly at the threshold: For example, how the speed of propagation of infection depends on subcritical p_a and what is the cluster size distribution for supercritical p_a.

Author response: We agree that further exploring the model away from the critical threshold is worthwhile. While our main focus has been on explaining the large degree of heterogeneity in outcomes – readily explained as a consequence of the sharp threshold-like behavior – we now include plots of the time-evolution of the infection (as well as the remaining states) over time for subcritical values of *pa*. The plots can be found in Figure S4 of the supplement.

**Reviewer #2 (Public Review):**
Xu et al. introduce a cellular automaton model to investigate the spatiotemporal spreading of viral infection. In this study, the author first analyzes the single-cell RNA sequencing data from experiments and identifies four clusters of cells at 48 hours post-viral infection, including susceptible cells (O), infected cells (V), IFN-secreting cells (N), and antiviral cells (A). Next, a cellular automaton model (NOVAa model) is introduced by assuming the existence of a transient pre-antiviral state (a). The model consists of an LxL lattice; each site represents one cell. The cells change their state following the rules depending on the interaction of neighboring cells. The model introduces a key parameter, p_a, representing the fraction of pre-antiviral state cells. Cell apoptosis is omitted in the model. Model simulations show a threshold-like behavior of the final attack rate of the virus when p_a changes continuously. There is a critical value p_c, so that when p_a < p_c, infections typically spread to the entire system, while at a higher p_a > p_c, the propagation of the infected state is inhibited. Moreover, the radius R that quantifies the diffusion range of N cells may affect the critical value p_c; a larger R yields a smaller value of the critical value p_c. The structure of clusters is different for different values of R; greater R leads to a different microscopic structure with fewer A and N cells in the final state. Compared with the single-cell RNA seq data, which implies a low fraction of IFN-positive cells - around 1.7% - the model simulation suggests R=5. The authors also explored a simplified version of the model, the OVA model, with only three states. The OVA model also has an outbreak size. The OVA model shows dynamics similar to the NOVAa model. However, the change in microstructure as a function of the IFN range R observed in the NOVAa model is not observed in the OVA model.

Author response: We thank the referee for the comprehensive summary of our work.

Data and model simulation mainly support the conclusions of this paper, but some weaknesses should be considered or clarified.

Author response: Thank you - we will address these point by point below.

(1) In the automaton model, the authors introduce a parameter p_a, representing the fraction of pre-antiviral state cells. The authors wrote: ``The parameter p_a can also be understood as the probability that an O cell will switch to the N or A state when exposed to the virus of IFNs, respectively.' Nevertheless, biologically, the fraction of pre-antiviral state cells does not mean the same value as the probability that an O cell switches to the N or A state. Moreover, in the numerical scheme, the cell state changes according to the deterministic role N(O)=a and N(a)=A. Hence, the probability p_a did not apply to the model simulation. It may need to clarify the exact meaning of the parameter p_a.

Author response: We acknowledge that this was an imprecise formulation, and have now changed it.

What we tried to convey with that comment was that, alternatively to having a certain fraction of cells be in the *a* state initially, one could instead have devised a model in which We should note that even the current model has a level of stochasticity, since we choose the cells to be updated with a constant probability rate - we choose N cells to update in each timestep, with replacement.

However, based on your suggestion, we simulated a version of the dynamics which included stochastic conversion, i.e. each action of a cell on a nearby cell happens only with a probability p_conv (and the original model is recovered as the p_conv=1 scenario). Of course, this slows down the dynamics (or effectively rescales time by a factor p_conv), but crucially we find that it does not appreciably affect the location of the threshold p_c. Below we include a parameter scan across p_a values for R=1 and p_conv=0.5, which shows that the threshold continues to appear at around p_a=27%. each O-state cell simply had a probability to *act* as an a-state cell upon exposure to the virus or to interferons, i.e. to switch to an N state (if exposed to virus) or to the A state (if exposed to interferons). In this simplified model, there would be no functional difference, since it would simply amount to whether each cell had a probability to be designated an a-cell initially (as in our model), or upon exposure. So our remark mainly served to explain that the role of the *p_a* parameter is simply to encode that a certain fraction of virus-naive cells behave this way (whether predetermined or not).

(2) The current model is deterministic. However, biologically, considering the probabilistic model may be more realistic. Are the results valid when the probability update strategy is considered? By the probability model, the cells change their state randomly to the state of the neighbor cells. The probability of cell state changes may be relevant for the threshold of p_a. It is interesting to know how the random response of cells may affect the main results and the critical value of p_a.

Author response: This is a good point - we are firm believers in the importance of stochasticity. We should note that even the current model has a level of stochasticity, since we choose the cells to be updated with a constant probability rate - we choose N cells to update in each timestep, with replacement.

However, based on your suggestion, we simulated a version of the dynamics which included *stochastic conversion*, i.e. each action of a cell on a nearby cell happens only with a probability p_conv (and the original model is recovered as the p_conv=1 scenario). Of course, this slows down the dynamics (or effectively rescales time by a factor p_conv), but crucially we find that it does not appreciably affect the location of the threshold p_c. Below we include a parameter scan across p_a values for R=1 and p_conv=0.5, which shows that the threshold continues to appear at around p_a=27%.

We now discuss these findings in the supplement and include the figure below as Fig. S5.

**Author response image 1. sa4fig1:** 

(3) Figure 2 shows a critical value p_c = 27.8% following a simulation on a lattice with dimension L = 1000. However, it is unclear if dimension changes may affect the critical value.

Author response: Re-running the simulations on a lattice 4x as large (i.e. L=2000) yields a similar critical value of 27-28% for *R=1,* so we are confident that finite size effects do not play a major role at L=1000 and beyond. For R=5, however, we find that a minimum lattice size greater than L=1000 is necessary to determine the critical threshold. Concretely, we find that the threshold value *pc* for R=5 changes somewhat when the lattice size is increased from 1000 to 2000, but is invariant under a change from 2000 to 3000, so we conclude that L=2000 is sufficient for *R=5*. The *pc* value for *R=5* cited in the manuscript (~0.4%) was determined from simulations at L=2000.

**Reviewer #3 (Public Review):**
Summary:This study considers how to model distinct host cell states that correspond to different stages of a viral infection: from naïve and susceptible cells to infected cells and a minority of important interferon-secreting cells that are the first line of defense against viral spread. The study first considers the distinct host cell states by analyzing previously published single-cell RNAseq data. Then an agent-based model on a square lattice is used to probe the dependence of the system on various parameters. Finally, a simplified version of the model is explored, and shown to have some similarity with the more complex model, yet lacks the dependence on the interferon range. By exploring these models one gains an intuitive understanding of the system, and the model may be used to generate hypotheses that could be tested experimentally, telling us "when to be surprised" if the biological system deviates from the model predictions.

Author response: Thank you for the summary! We agree with the role that you describe for a model such as this one.

Strengths:- Clear presentation of the experimental findings and a clear logical progression from these experimental findings to the modeling.- The modeling results are easy to understand, revealing interesting behavior and percolation-like features.- The scaling results presented span several decades and are therefore compelling. - The results presented suggest several interesting directions for theoretical follow-up work, as well as possible experiments to probe the system (e.g. by stimulating or blocking IFN secretion).Weaknesses:- Since the "range" of IFN is an important parameter, it makes sense to consider lattice geometries other than the square lattice, which is somewhat pathological. Perhaps a hexagonal lattice would generalize better.- Tissues are typically three-dimensional, not two-dimensional. (Epithelium is an exception). It would be interesting to see how the modeling translates to the three-dimensional case. Percolation transitions are known to be very sensitive to the dimensionality of the system.

Author response: We agree that probing different lattice geometries (2- and 3-dimensional alike) would be interesting and worthwhile. However, for this manuscript, we prefer to confine the analysis to the current, simple case. We do agree, however, that an extensive exploration of the role of geometry is an interesting future possibility.

- The fixed time-step of the agent-based modeling may introduce biases. I would consider simulating the system with Gillespie dynamics where the reaction rates depend on the ambient system parameters.- Single-cell RNAseq data typically involves data imputation due to the high sparsity of the measured gene expression. More information could be provided on this crucial data processing step since it may significantly alter the experimental findings.Justification of claims and conclusions:The claims and conclusions are well justified.
**Recommendations for the authors:**

**Reviewer #1 (Recommendations For The Authors):**
It is necessary to explain what UMAP does. Is clustering done in the space of twenty-something original dimensions or 2D? How UMAP1 and UMAP2 are selected and are those the same in all plots?

Author response: We have now added a few sentences to clarify the point raised above - the second snippet explains how clustering is performed:

“As a dimension reduction algorithm, UMAP is a manifold learning technique that favors the preservation of local distances over global distances (McInnes et al., 2018; Becht et al., 2019). It constructs a weighted graph from the data points and optimizes the graph layout in the low-dimensional space.”

“We cluster the cells with the principal components analysis (PCA) results from their gene expression. With the first 16 principal components, we calculate k-nearest neighbors and construct the shared nearest neighbor graph of the cells then optimize the modularity function to determine clusters. We present the cluster information on the UMAP plane and use the same UMAP coordinates for all the plots in this paper hereafter.”

Figure 1, what do bars in the upper right corners of panels d,e,f, and g indicate? ``Averaged' refers to time average? Something is missing in ``Cell proportions are labeled with corresponding colors in (a)' .

Author response: Thank you - we have now modified the figure caption. The bars in the upper right corners of panels d, e, f are color keys for gene expression, the brighter the color is, the higher the gene expression is.

“Averaged” gene expression refers to the mean expression of that particular gene across the cells within each indicated cluster.

The lines in (c) correspond to cell proportions in different states at different time points. The same state in (1) and (c) is shown in the same color.

Line 46, ``However' does not sound right in this context. Would ``Also' be better?

Author response: We agree and have corrected it in the revised manuscript.

Line 96``The viral genes are also partially expressed in these cells, but different from the 𝑁 cluster, the antiviral genes are fully expressed (Fig. S1 and S2).' The sentence needs to be rephrased.

Author response: We have rephrased the sentence: “As in the *N* cluster, the viral gene E is barely detected in these cells, indicating incomplete viral replication. However, in contrast to the *N* cluster, the antiviral genes are expressed to their full extent (Fig. S1 and S2).”

Line 126, missing "be", ``large' -> ``larger'.

Author response: Thank you, we have now corrected these typos.

Line 139-140 The logical link between ignoring apoptosis and the diffusion of IFN is unclear.

Author response: We modified the sentence as “Here, we assume that the secretion of IFNs by the 𝑁 cells is a faster process than possible apoptosis (Wen et al., 1997; Tesfaigzi, 2006) of these cells and that the diffusion of IFNs to the neighborhood is not significantly affected by apoptosis.”

Fig. 2a Do the yellow arrows show the effect of IFN and the purple arrows the propagation of viral infection?

Author response: That is correct. We have added this information to the figure caption: “The straight black arrows indicate transitions between cell states. The curved yellow arrows indicate the effects of IFNs on activating antiviral states. The curved purple arrows indicate viral spread to cells with 𝑂 and 𝑎 states.”

Fig. 3, n(s) as the axis label vs P(s) in the text? How do the curves in panel (a) look when the p_a is well above or below p_c?

Author response: Thank you. We have edited the labels in the figure to reflect the symbols used in the text.

Boundary conditions? From Fig. 4, apparently periodic?

Author response: Yes, we use periodic boundary conditions in the model. We clarify it in the model section now (last sentence).

It will be good to see a plot with time dependences of all cell types for a couple of values of p_a, illustrating propagation and cessation of the infection.

Author response: We agree, and have added a Figure S4 in the supplement which explores exactly that. Thank you for the suggestion.

A verbal qualitative description of why p_a has such importance and how the infection is terminated for large p_a would help.

**Reviewer #2 (Recommendations For The Authors):**
Below are two minor comments:(1) In the single-cell RNA sequencing data analysis, the authors describe the cell clusters O, V, A, and N. However, showing how the clusters are identified from the data might be more straightforward.

Author response: Technically, we cluster the cells using principal components analysis (PCA) results of their gene expression. With the first 16 principal components, we calculate *k*-nearest neighbors and construct the shared nearest neighbor graph of the cells and then optimize the modularity function to determine clusters. We manually annotate the clusters with O, V, A, and N based on the detected abundance of viral genes, antiviral genes, and IFNs.

(2) In Figure 3, what does n(s) mean in Figure 3a? And what is the meaning of the distribution P(s) of infection clusters? It may be stated clearly.

Author response: The use of *n(s)* was inconsistent, and we have now edited the figure to instead say *P(s)*, to harmonize it with the text. *P(s)* is the distribution of cluster sizes, *s*, expressed as a fraction of the whole system. In other words, once a cluster has reached its final size, we record *s=(N+V)/L^2* where *N* and *V* are the number of N and V state cells in the cluster (note that, by design, each simulation leads to a single cluster, since we seed the infection in one lattice point). We now indicate more clearly in the caption and the main text what exactly P(*s*) and *s* refer to.

**Reviewer #3 (Recommendations For The Authors):**
- Would the authors kindly share the simulation code with the community? Also, the data analysis code should be shared to follow current best practices. This needs to be standard practice in all publications. I would go as far as to say that in 2024 publishing a data analysis / simulation study without sharing the relevant code should be ostracized by the community.

Author response: We absolutely agree and have created a GitHub repository in which we share the C++ source code for the simulations and a Python notebook for plotting. The public repository can be found at https://github.com/BjarkeFN/ViralPercolation. We add this information in supplement under section “Code availability”.

- I would avoid the use of the wording "critical" threshold since this is almost guaranteed to infuriate a certain type of reader.

­

- Line 265 has a curious use of " ... " which should be replaced with something more appropriate.

Author response: Thank you for pointing it out! We have checked the typos.